# Removing Hidden Confounding by Experimental Grounding

**Nathan Kallus**
Cornell University and Cornell Tech
New York, NY
kallus@cornell.edu

**Aahlad Manas Puli**
New York University
New York, NY
apm470@nyu.edu

**Uri Shalit**
Technion
Haifa, Israel
urishalit@technion.ac.il

## Abstract

Observational data is increasingly used as a means for making individual-level causal predictions and intervention recommendations. The foremost challenge of causal inference from observational data is hidden confounding, whose presence cannot be tested in data and can invalidate any causal conclusion. Experimental data does not suffer from confounding but is usually limited in both scope and scale. We introduce a novel method of using limited experimental data to correct the hidden confounding in causal effect models trained on larger observational data, even if the observational data does not fully overlap with the experimental data. Our method makes strictly weaker assumptions than existing approaches, and we prove conditions under which it yields a consistent estimator. We demonstrate our method's efficacy using real-world data from a large educational experiment.

## 1 Introduction

In domains such as healthcare, education, and marketing there is growing interest in using observational data to draw causal conclusions about individual-level effects; for example, using electronic healthcare records to determine which patients should get what treatments, using school records to optimize educational policy interventions, or using past advertising campaign data to refine targeting and maximize lift. Observational datasets, due to their often very large number of samples and exhaustive scope (many measured covariates) in comparison to experimental datasets, offer a unique opportunity to uncover fine-grained effects that may apply to many target populations.

However, a significant obstacle when attempting to draw causal conclusions from observational data is the problem of *hidden confounders*: factors that affect both treatment assignment and outcome, but are unmeasured in the observational data. Example cases where hidden confounders arise include physicians prescribing medication based on indicators not present in the health record, or classes being assigned a teacher's aide because of special efforts by a competent school principal. Hidden confounding can lead to no-vanishing bias in causal estimates even in the limit of infinite samples [Pea09].

In an observational study, one can never prove that there is no hidden confounding [Pea09]. However, a possible fix can be found if there exists a Randomized Controlled Trial (RCT) testing the effect of the intervention in question. For example, if a Health Management Organization (HMO) is considering the effect of a medication on its patient population, it might look at an RCT which tested this medication. The problem with using RCTs is that often their participants do not fully reflect the target population. As an example, an HMO in California might have to use an RCT from Switzerland, conducted perhaps several years ago, on a much smaller population. The problem of generalizing conclusions from an RCT to a different target population is known as the problem of external validity [Rot05, AO17], or more specifically, transportability [BP13, PB14].

In this paper, we are interested in the case where fine-grained causal inference is sought, in the form of Conditional Average Treatment Effects (CATE), where we consider a large set of covariates, enough to identify each unit. We aim at using a large observational sample and a possibly much smaller experimental sample. The typical use case we have in mind is of a user who wishes to estimate CATE and has a relatively large observational sample that covers their population of interest. This observational sample might suffer from hidden confounding, as all observational data will to some extent, but they also have a smaller sample from an experiment, albeit one that might not directly reflect their population of interest. For example, consider The Women's Health Initiative [Ros02] where there was a big previous observational study and a smaller RCT to study hormone replacement therapy. The studies ended up with opposite results and there is intense discussion about confounding and external validity: the RCT was limited due to covering a fundamentally different (healthier and younger) population compared with the observational study [HAL$^+$08, Van09].

Differently from previous work on estimating CATE from observational data, our approach *does not* assume that all confounders have been measured, and we only assume that the support of the experimental study has some overlap with the support of the observational study. The major assumption we do make is that we can learn the *structure* of the hidden confounding by comparing the observational and experimental samples. Specifically, rather than assuming that effects themselves have a parametric structure – a questionable assumption that is bound to lead to dangerous extrapolation from small experiments – we only assume that this hidden confounding function has a parametric structure that we can extrapolate. Thus we limit ourselves to a parametric correction of a possibly complex effect function learned on the observational data. We discuss why this assumption is possibly reasonable. Specifically, as long as the parametric family includes the zero function, this assumption is strictly weaker than assuming that all confounders in the observational study have been observed. One way to view our approach is that we bring together an unbiased but high-variance estimator from the RCT (possibly infinite-variance when the RCT has zero overlap with the target population) and a biased but low-variance estimator from the observational study. This achieves a consistent (vanishing bias *and* variance) CATE estimator. Finally, we run experiments on both simulation and real-world data and show our method outperforms the standard approaches to this problem. In particular, we use data from a large-scale RCT measuring the effect of small classrooms and teacher's aids [WJB$^+$90, Kru99] to obtain ground-truth estimates of causal effects, which we then try and reproduce from a confounded observational study.

## 2  Setup

We focus on studying a binary treatment, which we interpret as the presence or absence of an intervention of interest. To study its fine-grained effects on individuals, we consider having treatment-outcome data from two sources: an observational study that may be subject to hidden confounding, and an unconfounded study, typically coming from an experiment. The observational data consists of baseline covariates $X_i^{\mathrm{Conf}} \in \mathbb{R}^d$, assigned treatments $T_i^{\mathrm{Conf}} \in \{0, 1\}$, and observed outcomes $Y_i^{\mathrm{Conf}} \in \mathbb{R}$ for $i = 1, \ldots, n^{\mathrm{Conf}}$. Similarly, the unconfounded data consists of $X_i^{\mathrm{Unc}}, T_i^{\mathrm{Unc}}, Y_i^{\mathrm{Unc}}$ for $i = 1, \ldots, n^{\mathrm{Unc}}$.

Conceptually, we focus on the setting where (1) the observational data is of much larger scale $n^{\mathrm{Unc}} \ll n^{\mathrm{Conf}}$ and/or (2) the support of the unconfounded data $\mathrm{Support}(X_i^{\mathrm{Unc}}) = \{x : \mathbb{P}\left(\|X_i^{\mathrm{Unc}} - x\| \le \delta\right) > 0 \ \forall \delta > 0\}$, does not include the population about which we want to make causal conclusions and targeted interventions. This means that the observational data has both the scale and the scope we want but the presence of confounding limits the study of causal effects, while the unconfounded experimental data has unconfoundedness but does not have the scale and/or scope necessary to study the individual-level effects of interest.

The unconfounded data usually comes from an RCT that was conducted on a smaller scale on a different population, as presented in the previous section. Alternatively, and equivalently for our formalism, it can arise from recognizing a latent unconfounded sub-experiment within the observational study. For example, we may have information from the data generation process that indicates that treatment for certain units was actually assigned purely as a (possibly stochastic) function of the observed covariates $x$. Two examples of this would be when certain prognoses dictate a strict rule-based treatment assignment or in situations of known equipoise after a certain prognosis, where there is no evidence guiding treatment one way or the other and its assignment is as if at random based on the individual who ends up administering it. Regardless if the unconfounded data came

from a secondary RCT (more common) or from within the observational dataset, our mathematical set up remains the same.

Formally, we consider each dataset to be iid draws from two different super-populations, indicated by the event $E$ taking either the value $E^{\mathrm{Conf}}$ or $E^{\mathrm{Unc}}$. The observational data are iid draws from the population given by conditioning on the event $E^{\mathrm{Conf}}$: $X_i^{\mathrm{Conf}}, T_i^{\mathrm{Conf}}, Y_i^{\mathrm{Conf}} \sim (X, T, Y \mid E^{\mathrm{Conf}})$ iid. Similarly, $X_i^{\mathrm{Unc}}, T_i^{\mathrm{Unc}}, Y_i^{\mathrm{Unc}} \sim (X, T, Y \mid E^{\mathrm{Unc}})$. Using potential outcome notation, assuming the standard Stable Unit Treatment Value Assumption (SUTVA), which posits no interference and consistency between observed and potential outcomes, we let $Y(0), Y(1)$ be the potential outcomes of administering each of the two treatments and $Y = Y(T) = TY(1) + (1 - T)Y(0)$. The quantity we are interested in is the Conditional Average Treatment Effect (CATE):

**Definition 1** (CATE). Let $\tau(x) = \mathbb{E}\left[Y(1) - Y(0) | X = x\right]$.

The key assumption we make about the unconfounded data is its unconfoundedness:

**Assumption 1.** [Unconfounded experiment]

$$(i) \ \ Y(0), Y(1) \perp\!\!\!\perp T \mid X, \ E^{\mathrm{Unc}}$$
$$(ii) \ \ Y(0), Y(1) \perp\!\!\!\perp E^{\mathrm{Unc}} \mid X.$$

This assumption holds if the unconfounded data was generated in a randomized control trial. More generally, it is functionally equivalent to assuming that the unconfounded data was generated by running a logging policy on a contextual bandit, that is, first covariates are drawn from the unconfounded population $X \mid E^{\mathrm{Unc}}$ and revealed, then a treatment $T$ is chosen, the outcomes are drawn based on the covariates $Y(0), Y(1) \mid X$, but only the outcome corresponding to the chosen treatment $Y = Y(T)$ is revealed. The second part of the assumption means that merely being in the unconfounded study does not affect the potential outcomes conditioned on the covariates $X$. It implies that the functional relationship between the unobserved confounders and the potential outcomes is the same in both studies. This will fail if for example knowing you are part of a study causes you to react differently to the same treatment. We note that this assumption is strictly weaker than the standard ignorability assumption in observational studies. This assumption implies that for covariates *within the domain of the experiment*, we can identify the value of CATE using regression. Specifically, if $x \in \mathrm{Support}(X \mid E^{\mathrm{Unc}})$, that is, if $\mathbb{P}\left(\|X - x\| \leq \delta \mid E^{\mathrm{Unc}}\right) > 0 \ \forall \delta > 0$, then $\tau(x) = \mathbb{E}\left[Y \mid T = 1, X = x, E^{\mathrm{Unc}}\right] - \mathbb{E}\left[Y \mid T = 0, X = x, E^{\mathrm{Unc}}\right]$, where $\mathbb{E}\left[Y \mid T = t, X = x, E^{\mathrm{Unc}}\right]$ can be identified by regressing observed outcomes on treatment and covariates in the unconfounded data. However, this identification of CATE is (i) limited to the restricted domain of the experiment and (ii) hindered by the limited amount of data available in the unconfounded sample. The hope is to overcome these obstacles using the observational data.

Importantly, however, the unconfoundedness assumption is *not* assumed to hold for the observational data, which may be subject to unmeasured confounding. That is, *both* selection into the observational study *and* the selection of the treatment may be confounded with the potential outcomes of any one treatment. Let us denote the difference in conditional average outcomes in the observational data by

$$\omega(x) = \mathbb{E}\left[Y \mid T = 1, X = x, E^{\mathrm{Conf}}\right] - \mathbb{E}\left[Y \mid T = 0, X = x, E^{\mathrm{Conf}}\right].$$

Note that due to confounding factors, $\omega(x) \neq \tau(x)$ for *any* $x$, whether in the support of the observational study or not. The difference between these two quantities is precisely the confounding effect, which we denote

$$\eta(x) = \tau(x) - \omega(x).$$

Another way to express this term is:

$$\eta(x) = \{\mathbb{E}\left[Y(1)|x\right] - \mathbb{E}\left[Y(1)|x, T = 1\right]\} - \{\mathbb{E}\left[Y(0)|x\right] - \mathbb{E}\left[Y(0)|x, T = 0\right]\}.$$

Note that if the observational study were unconfounded then we would have $\eta(x) = 0$. Further note that a standard assumption in the vast majority of methodological literature makes the assumption that $\eta(x) \equiv 0$, even though it is widely acknowledged that this assumption isn't realistic, and is at best an approximation.

**Example.** In order to better understand the function $\eta(x)$, consider the following case: Assume there are two equally likely types of patients, "dutiful" and "negligent". Dutiful patients take care of their general health and are more likely to seek treatment, while negligent patients do not. Assume

$T = 1$ is a medical treatment that requires the patient to see a physician, do lab tests, and obtain a prescription if indeed needed, while $T = 0$ means no treatment. Let $Y$ be some measure of health, say blood pressure. In this scenario, where patients are self-selected into treatment (to a certain degree), we would expect that both potential outcomes would be greater for the treated over the control: $\mathbb{E}\left[Y(1)|T = 1\right] > \mathbb{E}\left[Y(1)|T = 0\right]$, $\mathbb{E}\left[Y(0)|T = 1\right] > \mathbb{E}\left[Y(0)|T = 0\right]$. Since $\mathbb{E}\left[Y(1)\right] = \mathbb{E}\left[Y(1)|T = 1\right] p(T = 1) + \mathbb{E}\left[Y(1)|T = 0\right] p(T = 0)$ we also have that $\mathbb{E}\left[Y(1)\right] > \mathbb{E}\left[Y(1)|T = 1\right]$, and $\mathbb{E}\left[Y(0)\right] > \mathbb{E}\left[Y(0)|T = 0\right]$ unless $p(T = 0) = 1$. Taken together, this shows that in the above scenario, we expect $\eta < 0$, if we haven't measured any $X$. This logic carries through in the plausible scenario where we have measured some $X$, but do not have access to all the variables $X$ that allows us to tell apart "dutiful" from "negligent" patients. To sum up, this example shows that in cases where some units are selected such as those more likely to be treated are those whose potential outcomes are higher (resp. lower) anyway, we can expect $\eta$ to be negative (resp. positive).

## 3 Method

Given data from both the unconfounded and confounded studies, we propose the following recipe for removing the hidden confounding. First, we learn a function $\hat{\omega}$ over the observational sample $\{X_i^{\text{Conf}}, T_i^{\text{Conf}}, Y_i^{\text{Conf}}\}_{i=1}^{n^{\text{Conf}}}$. This can be done using any CATE estimation method such as learning two regression functions for the treated and control and taking their difference, or specially constructed methods such as Causal Forest [WA17]. Since we assume this sample has hidden confounding, $\omega$ is not equal to the true CATE and correspondingly $\hat{\omega}$ does not estimate the true CATE. We then learn a correction term which interpolates between $\hat{\omega}$ evaluated on the RCT samples $X_i^{\text{Unc}}$, and the RCT outcomes $Y_i^{\text{Unc}}$. This is a correction term for hidden confounding, which is our estimate of $\eta$. The correction term allows us to extrapolate $\tau$ over the confounded sample, using the identity $\tau(X) = \omega(X) + \eta(X)$.

Note that we could not have gone the other way round: if we were to start with estimating $\tau$ over the unconfounded sample, and then estimate $\eta$ using the samples from the confounded study, we would end up constructing an estimate of $\omega(x)$, which is not the quantity of interest. Moreover, doing so would be difficult as the unconfounded sample is not expected to cover the confounded one.

Specifically, the way we use the RCT samples relies on a simple identity. Let $e^{\text{Unc}}(x) = \mathbb{P}\left(T = 1 \mid X = x, E^{\text{Unc}}\right)$ be the propensity score on the unconfounded sample. If this sample is an RCT then typically $e^{\text{Unc}}(x) = q$ for some constant, often $q = 0.5$.

Let $q(X_i^{\text{Unc}}) = \frac{T_i^{\text{Unc}}}{e^{\text{Unc}}(X_i^{\text{Unc}})} - \frac{1 - T_i^{\text{Unc}}}{1 - e^{\text{Unc}}(X_i^{\text{Unc}})}$ be a signed re-weighting function. We have:

**Lemma 1.**

$$\mathbb{E}\left[q(X_i^{Unc})Y_i^{Unc}|X_i^{Unc}, E^{Unc}\right] = \tau(X_i^{Unc}). \tag{1}$$

What Lemma 1 shows us is that $q(X_i^{\text{Unc}})Y_i^{\text{Unc}}$ is an unbiased estimate of $\tau(X_i^{\text{Unc}})$. We now use this fact to learn $\eta$ as follows:

$$\hat{\theta} = \arg\min_\theta \sum_{i=1}^{n^{\text{Unc}}} \left(q(X_i^{\text{Unc}})Y_i^{\text{Unc}} - \hat{\omega}(X_i^{\text{Unc}}) - \theta^\top X_i^{\text{Unc}}\right)^2 \tag{2}$$

Let

$$\hat{\tau}(x) = \hat{\omega}(x) + \hat{\theta}^\top x. \tag{3}$$

The method is summarized in Algorithm 1.

Let us contrast our approach with two existing ones. The first, is to simply learn the treatment effect function directly from the unconfounded data, and extrapolate it to the observational sample. This is guaranteed to be unconfounded, and with a large enough unconfounded sample the CATE function can be learned [CHIM08, Pea15]. This approach is presented for example by [BP13] for ATE, as the transport formula. However, extending this approach to CATE in our case is not as straightforward. The reason is that we assume that the confounded study does not fully overlap with the unconfounded study, which requires *extrapolating* the estimated CATE function into a region of sample space outside the region where it was fit. This requires strong parametric assumptions about the CATE

---

**Algorithm 1** Remove hidden confounding with unconfounded sample

---

1: **Input:** Unconfounded sample with propensity scores $D^{\text{Unc}} = \{X_i^{\text{Unc}}, T_i^{\text{Unc}}, Y_i^{\text{Unc}}, e^{\text{Unc}}(X_i^{\text{Unc}})\}_{i=1}^{n^{\text{Unc}}}$. Confounded sample $D^{\text{Conf}} = \{X_i^{\text{Conf}}, T_i^{\text{Conf}}, Y_i^{\text{Conf}}\}_{i=1}^{n^{\text{Conf}}}$. Algorithm $\mathcal{Q}$ for fitting CATE.
2: Run $\mathcal{Q}$ on $D^{\text{Conf}}$, obtain CATE estimate $\hat{\omega}$.
3: Let $\hat{\theta}$ be the solution of the optimization problem in Equation (2).
4: Set function $\hat{\tau}(x) := \hat{\omega}(x) + \hat{\theta}^\top x$
5: **Return:** $\hat{\tau}$, an estimate of CATE over $D^{\text{Conf}}$.

---

function. On the other hand, we do have samples from the target region, they are simply confounded. One way to view our approach is that we move the extrapolation a step back: instead of extrapolating the CATE function, we merely extrapolate a correction due to hidden confounding. In the case that the CATE function does actually extrapolate well, we do no harm - we learn $\eta \approx 0$.

The second alternative relies on re-weighting the RCT population so as to make it similar to the target, observational population [SCBL11, HGRS15, AO17]. These approaches suffer from two important drawbacks from our point of view: (i) they assume the observational study has no unmeasured confounders, which is often an unrealistic assumption; (ii) they assume that the support of the observational study is contained within the support of the experimental study, which again is unrealistic as the experimental studies are often smaller and on somewhat different populations. If we were to apply these approaches to our case, we would be re-weighting by the inverse of weights which are close to, or even identical to, 0.

## 4 Theoretical guarantee

We prove that under conditions of parametric identification of $\eta$, Algorithm 1 recovers a consistent estimate of $\tau(x)$ over the $\mathbb{E}^{\text{Conf}}$, at a rate which is governed by the rate of estimating $\omega$ by $\hat{\omega}$. For the sake of clarity, we focus on a linear specification of $\eta$. Other parametric specifications can easily be accommodated given that the appropriate identification criteria hold (for linear this is the non-singularity of the design matrix). Note that this result is strictly stronger than results about CATE identification which rely on ignorability: what enables the improvement is of course the presence of the unconfounded sample $E^{\text{Unc}}$. Also note that this result is strictly stronger than the transport formula [BP13] and re-weighting such as [AO17].

**Theorem 1.** *Suppose*

1. *$\hat{\omega}$ is a consistent estimator on the observational data (on which it's trained): $\mathbb{E}[(\hat{\omega}(X) - \omega(X))^2 \mid E^{Conf}] = O(r(n))$ for $r(n) = o(1)$*

2. *The covariates in the confounded data cover those in the unconfounded data (strong one-way overlap): $\exists \kappa > 0 : \mathbb{P}\left(E^{Unc} \mid X\right) \leq \kappa \mathbb{P}\left(E^{Conf} \mid X\right)$*

3. *$\eta$ is linear: $\exists \theta_0 : \eta(x) = \theta_0^\top x$*

4. *Identifiability of $\theta_0$: $\mathbb{E}[XX^\top \mid E^{Unc}]$ is non-singular*

5. *$X, Y$, and $\hat{\omega}(X)$ have finite fourth moments in the experimental data: $\mathbb{E}[\|X\|_2^4 \mid E^{Unc}] < \infty$, $\mathbb{E}[Y^4 \mid E^{Unc}] < \infty$, $\mathbb{E}[\hat{\omega}(X)^4 \mid E^{Unc}] < \infty$*

6. *Strong overlap between treatments in unconfounded data: $\exists \nu > 0 : \nu \leq e^{Unc}(X) \leq 1 - \nu$*

*Then $\hat{\theta}$ is consistent*

$$\|\hat{\theta} - \theta_0\|_2^2 = O_p(r(n) + 1/n)$$

*and $\hat{\tau}$ is consistent on its target population*

$$((\hat{\tau}(X) - \tau(X))^2 \mid E^{Conf}) = O_p(r(n) + 1/n)$$

There are a few things to note about the result and its conditions. First, we note that if the so-called confounded observational sample is in fact unconfounded then we immediately get that the linear

specification of $\eta$ is correct with $\theta_0 = 0$ because we simply have $\eta(x) = 0$. Therefore, our conditions are strictly weaker than imposing unconfoundedness on the observational data.

Condition 1 requires that our base method for learning $\omega$ is consistent just as a regression method. There are a few ways to guarantee this. For example, if we fit $\hat{\omega}$ by empirical risk minimization on weighted outcomes over a function class of finite capacity (such as a VC class) or if we fit as the difference of two regression functions each fit by empirical risk minimization on observed outcomes in each treatment group, then standard results in statistical learning [BM02] ensure the consistency of L2 risk and therefore the L2 convergence required in condition 1. Alternatively, any method for learning CATE that would have been consistent for CATE under unconfoundedness would actually still be consistent for $\omega$ if applied. Therefore we can also rely on such base method as causal forests [WA17] and other methods that target CATE as inputs to our method, even if they don't actually learn CATE here due to confounding.

Condition 2 captures our understanding of the observational dataset having a larger scope than the experimental dataset. The condition essentially requires a strong form of absolute continuity between the two covariate distributions. This condition could potentially be relaxed so long as there is enough intersection where we can learn $\eta$. So for example, if there is a subset of the experiment that the observational data covers, that would be sufficient so long as we can also ensure that condition 4 still remains valid on that subset so that we can learn the sufficient parameters for $\eta$.

Condition 3, the linear specification of $\eta$, can be replaced with another one so long as it has finitely many parameters and they can be identified on the experimental dataset, i.e., condition 4 above would change appropriately.

Since unconfoundedness implies $\eta = 0$, whenever the parametric specification of $\eta$ contains the zero function (e.g., as in the linear case above since $\theta_0 = 0$ is allowed) condition 3 is *strictly weaker* than assuming unconfoundedness. In that sense, our method can consistently estimate CATE on a population where no experimental data exists under weaker conditions than existing methods, which assume the observational data is unconfounded.

Condition 5 is trivially satisfied whenever outcomes and covariates are bounded. Similarly, we would expect that if the first two parts of condition 5 hold (about $X$ and $Y$) then the last one about $\hat{\omega}$ would also hold as it is predicting outcomes $Y$. That is, the last part of condition 5 is essentially a requirement on our $\hat{\omega}$-leaner base method that it's not doing anything strange like *adding* unnecessary noise to $Y$ thereby making it have fewer moments. For all base methods that we consider, this would come for free because they are only averaging outcomes $Y$. We also note that if we impose the existence of even higher moments as well as pointwise asymptotic normality of $\hat{\omega}$, one can easily transform the result to an asymptotic normality result. Standard error estimates will in turn require a variance estimate of $\hat{\omega}$.

Finally, we note that condition 6, which requires strong overlap, only needs to hold in the *unconfounded* sample. This is important as it would be a rather strong requirement in the confounded sample where treatment choices may depend on high dimensional variables [DDF$^+$17], but it is a weak condition for the experimental data. Specifically, if the unconfounded sample arose from an RCT then propensities would be constant and the condition would hold trivially.

## 5 Experiments

In order to illustrate the validity and usefulness of our proposed method we conduct simulation experiments and experiments with real-world data taken from the Tennessee STAR study: a large long-term school study where students were randomized to different types of classes [WJB$^+$90, Kru99].

### 5.1 Simulation study

We generate data simulating a situation where there exists an un-confounded dataset and a confounded dataset, with only partial overlap. Let $X \in \mathbb{R}$ be a measured covariate, $T \in \{0, 1\}$ binary treatment assignment, $U \in \mathbb{R}$ an unmeasured confounder, and $Y \in \mathbb{R}$ the outcome. We are interested in $\tau(X)$.

We generate the unconfounded sample as follows: $X^{\mathrm{Unc}} \sim$ Uniform $[-1, 1]$, $U^{\mathrm{Unc}} \sim \mathcal{N}(0, 1)$, $T^{\mathrm{Unc}} \sim$ Bernoulli$(0.5)$. We generate the confounded sample as follows: we first sample $T^{\mathrm{Conf}} \sim$

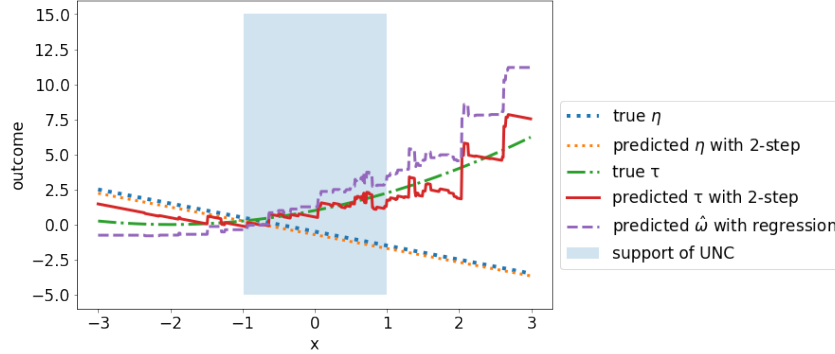

**Figure 1:** *True and predicted $\tau$ and $\eta$ for an unconfounded sample of limited overlap with the confounded sample: unconfounded samples are limited to $[-1, 1]$ (blue shaded region); confounded samples lie in $-[3, 3]$; predicted $\hat{w}$ is from difference of regressions on treated and control $y$.*

Bernoulli$(0.5)$ and then sample $X^{\text{Conf}}, U^{\text{Conf}}$ from a bivariate Gaussian

$$(X^{\text{Conf}}, U^{\text{Conf}})|T^{\text{Conf}} \sim \mathcal{N}\left([0, 0], \begin{bmatrix} 1 & T^{\text{Conf}} - 0.5 \\ T^{\text{Conf}} - 0.5 & 1 \end{bmatrix}\right).$$

This means that $X^{\text{Conf}}, U^{\text{Conf}}$ come from a Gaussian mixture model where $T^{\text{Conf}}$ denotes the mixture component and the components have equal means but different covariance structures. This also implies that $\eta$ is linear.

For both datasets we set $Y = 1 + T + X + 2 \cdot T \cdot X + 0.5X^2 + 0.75 \cdot T \cdot X^2 + U + 0.5\epsilon$, where $\epsilon \sim \mathcal{N}(0, 1)$. The true CATE is therefore $\tau(X) = 0.75X^2 + 2X + 1$. We have that the true $\omega = \tau + \mathbb{E}[U|X, T = 1] - \mathbb{E}[U|X, T = 0]$, which leads to the true $\eta = x$. We then apply our method (with a CF base) to learn $\eta$. We plot (See Figure 1) here the true and recovered $\eta$ with our method. Even with the limited un-confounded set (between $-1, 1$) making the full scope of the $x^2$ term in $Y$ inaccessible, we are able to reasonably estimate $\tau$. Other methods would suffer under the strong unobserved confounding.

## 5.2 Real-world data

Validating causal-inference methods is hard because we almost never have access to true counterfactuals. We approach this challenge by using data from a randomized controlled trial, the Tennessee STAR study [WJB+90, Kru99, MISN18]. When using an RCT, we have access to unbiased CATE-estimates because we are guaranteed unconfoundedness. We then artificially introduce confounding by selectively removing a biased subset of samples.

**The data:** The Tennessee Student/Teacher Achievement Ratio (STAR) experiment is a randomized experiment started in 1985 to measure the effect of class size on student outcomes, measured by standardized test scores. The experiment started monitoring students in kindergarten and followed students until third grade. Students and teachers were randomly assigned into conditions during the first school year, with the intention for students to continue in their class-size condition for the entirety of the experiment. We focus on two of the experiment conditions: small classes(13-17 pupils), and regular classes(22-25 pupils). Since many students only started the study at first grade, we took as treatment their class-type at first grade. Overall we have 4509 students with treatment assignment at first grade. The outcome $Y$ is the sum of the listening, reading, and math standardized test at the end of first grade. After removing students with missing outcomes [1], we remain with a randomized sample of 4218 students: 1805 assigned to treatment (small class, $T = 1$), and 2413 to control (regular size class, $T = 0$). In addition to treatment and outcome, we used the following covariates for each student: gender, race, birth month, birthday, birth year, free lunch given or not, teacher id. Our goal is to compute the CATE conditioned on this set of covariates, jointly denoted $X$.

**Computing ground-truth CATE:** The STAR RCT allows us to obtain an unbiased estimate of the CATE. Specifically, we use the identity in Eq. (1), and the fact that in the study, the propensity scores

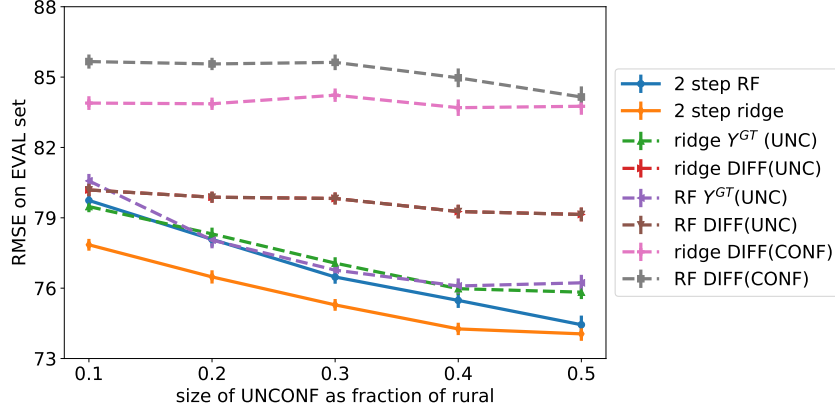

**Figure 2:** *RMSE of estimating $Y_i^{GT}$ on a held-out evaluation subset of ALL \ UNC, for varying sizes of the unconfounded subset.* **RF** *and* ridge *stand for Random Forest and Ridge Regression, respectively.* **2 step** *is our method.* **[RF/ridge] $Y_i^{GT}$** *is regression directly on $Y_i^{GT}$.* **[RF/ridge] DIFF** *is the difference between the predictions of models fit on treated and control separately.* **UNC** *or* **CONF** *in parentheses indicates which subset of the data was used for regression.*

$e(X_i)$ were constant. We define a ground-truth sample $\{(X_i, Y_i^{GT})\}_{i=1}^n$, where $Y_i^{GT} = \frac{Y_i}{q+T_i-1}$, $q = p(T = 1)$. By Eq. (1) we know that $\mathbb{E}\left[Y_i^{GT}|X_i\right] = \tau(X_i)$ within the STAR study.

**Introducing hidden confounding:** Now that we have the ground-truth CATE, we wish to emulate the scenario which motivates our work. We split the entire dataset (ALL) into a small unconfounded subset (UNC), and a larger, confounded subset (CONF) over a somewhat different population. We do this by splitting the population over a variable which is known to be a strong determinant of outcome [Kru99]: rural or inner-city (2811 students) vs. urban or suburban (1407 students).

We generate UNC by randomly sampling a fraction $q'$ of the rural or inner-city students, where $q'$ ranges from $0.1$ to $0.5$. Over this sample, we know that treatment assignment was at random.

When generating CONF, we wish to obtain two goals: (a) the support of CONF should have only a partial overlap with the support of UNC, and (b) treatment assignment should be confounded, i.e. the treated and control populations should be systematically different in their potential outcomes. In order to achieve these goals, we generate CONF as follows: From the rural or inner-city students, we take the controls ($T = 0$) that were not sampled in UNC, and *only* the treated ($T = 1$) whose outcomes were in the lower half of outcomes among treated rural or inner-city students. From the urban or suburban students, we take all of the controls, and *only* the treated whose outcomes were in the lower half of outcomes among treated urban or suburban students.

This procedure results in UNC and CONF populations which do not fully overlap: UNC has only rural or inner-city students, while CONF has a substantial subset (roughly one half for $q' = 0.5$) of urban and suburban students. It also creates confounding, by removing the students with the higher scores selectively from the treated population. This biases the naive treatment effect estimates downward. We further complicate matters by dropping the covariate indicating rural, inner-city, urban or suburban from all subsequent analysis. Therefore, we have significant unmeasured confounding in the CONF population, and also the unconfounded ground-truth in the original, ALL population.

**Metric:** In our experiments, we assume we have access to samples from UNC and CONF. We use either UNC, CONF or both to fit various models for predicting CATE. We then evaluate how well the CATE predictions match $Y_i^{GT}$ on a held-out sample from ALL \ UNC (the set ALL minus the set UNC), in terms of RMSE. Note that we are *not* evaluating on CONF, but on the unconfounded version of CONF, which is exactly ALL \ UNC. The reason we don't evaluate on ALL is twofold: First, it will only make the task easier because of the nature of the UNC set; second, we are motivated by the scenario where we have a confounded observational study representing the target population of interest, and wish to be aided by a separate unconfounded study (typically an RCT) available for a different population. We focus on a held-out set in order to avoid giving too much of an advantage to methods which can simply fit the observed outcomes well.

**Baselines:** As a baseline we fit CATE using standard methods on either the UNC set or the CONF set. Fitting on the UNC set is essentially a CATE version of applying the transport formula [PB14]. Fitting on the CONF set amounts to assuming ignorability (which is wrong in this case), and using standard methods. The methods we use to estimate CATE are: (i) Regression method fit on $Y_i^{GT}$ over UNC (ii) Regression method fit separately on treated and control in CONF (iii) Regression method fit separately on treated and control in UNC. The regression methods we use in (i)-(iii) are Random Forest with 200 trees and Ridge Regression with cross-validation. In baselines (ii) and (iii), the CATE is estimated as the difference between the prediction of the model fit on the treated and the prediction of the model fit on the control. We also experimented extensively with Causal Forest [WA17], but found it to uniformly perform worse than the other methods, even when given unfair advantages such as access to the entire dataset (ALL).

**Results:** Our two-step method requires a method for fitting $\hat{\omega}$ on the confounded dataset. We experiment with two methods, which parallel those used as baseline: A regression method fit separately on treated and control in CONF, where we use either Random Forest with 200 trees or Ridge Regression with cross-validation as regression methods. We see that our methods, *2-step RF* and *2-step ridge*, consistently produce more accurate estimates than the baselines. We see that our methods in particular are able to make use of larger unconfounded sets to produce better estimates of the CATE function.See Figure 2 for the performance of our method vs. the various baselines.

## 6   Discussion

In this paper we address a scenario that is becoming more and more common: users with large observational datasets who wish to extract causal insights using their data and help from unconfounded experiments on different populations. One direction for future work is combining the current work with work that looks explicitly into the causal graph connecting the covariates, including unmeasured ones [TT15, MMC16]. Another direction includes cases where the outcomes or interventions are not directly comparable, but where the difference can be modeled. For example, experimental studies often only study short-term outcomes, whereas the observational study might track long-term outcomes which are of more interest [ACIK16].

## Acknowledgements

We wish to thank the anonymous reviewers for their helpful suggestions and comments. (NK) This material is based upon work supported by the National Science Foundation under Grant No. 1656996.

## Footnotes

[1]The correlation between missing outcome and treatment assignment is $R^2 < 10^{-4}$.

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
