[Supplementary Material · rhceg-cr-supplementary.pdf]

# A Proofs

*Proof of Lemma 1.* We wish to prove $\mathbb{E}\left[q(X_i^{\text{Unc}})Y_i^{\text{Unc}}|X_i^{\text{Unc}}, E^{\text{Unc}}\right] = \tau(X_i^{\text{Unc}})$. We have that:

$$\mathbb{E}\left[q(X_i^{\text{Unc}})Y_i^{\text{Unc}}|X_i^{\text{Unc}}, E^{\text{Unc}}\right] = \mathbb{E}\left[q(X_i^{\text{Unc}})Y_i^{\text{Unc}}|X_i^{\text{Unc}}, E^{\text{Unc}}, T = 1\right] e^{\text{Unc}}(X_i^{\text{Unc}}) \tag{3}$$
$$+ \mathbb{E}\left[q(X_i^{\text{Unc}})Y_i^{\text{Unc}}|X_i^{\text{Unc}}, E^{\text{Unc}}, T = 0\right]\left(1 - e^{\text{Unc}}(X_i^{\text{Unc}})\right) \tag{4}$$
$$= \mathbb{E}\left[Y_i^{\text{Unc}}|X_i^{\text{Unc}}, E^{\text{Unc}}, T = 1\right] - \mathbb{E}\left[Y_i^{\text{Unc}}|X_i^{\text{Unc}}, E^{\text{Unc}}, T = 0\right] \tag{5}$$
$$\overset{(a)}{=} \mathbb{E}\left[Y(1)|X_i^{\text{Unc}}\right] - \mathbb{E}\left[Y(0)|X_i^{\text{Unc}}\right] = \tau(X_i^{\text{Unc}}),$$

where equality (a) is by Assumption 1 and the definition of $Y$.

$\square$

*Proof of Thm. 1.* Let $\mathbf{X}^{\text{Unc}} = (X_1^{\text{Unc}}, \ldots, X_{n^{\text{Unc}}}^{\text{Unc}})$ be the design matrix in the experimental data and let $u_i^{\text{Unc}} = \left(\frac{T_i^{\text{Unc}}}{e^{\text{Unc}}(X_i^{\text{Unc}})} - \frac{1-T_i^{\text{Unc}}}{1-e^{\text{Unc}}(X_i^{\text{Unc}})}\right)Y_i^{\text{Unc}} - \hat{\omega}(X_i^{\text{Unc}})$ be the regression outcome and $\mathbf{u} = (u_1^{\text{Unc}}, \ldots, u_{n^{\text{Unc}}}^{\text{Unc}})$ so that $\hat{\theta} = (\mathbf{X}^{\text{Unc}}(\mathbf{X}^{\text{Unc}})^\top)^{-1}(\mathbf{X}^{\text{Unc}})^\top \mathbf{u}^{\text{Unc}}$. Let $a_i^{\text{Unc}} = \omega(X_i^{\text{Unc}}) - \hat{\omega}(X_i^{\text{Unc}})$ and $b_i^{\text{Unc}} = \left(\frac{T_i^{\text{Unc}}}{e^{\text{Unc}}(X_i^{\text{Unc}})} - \frac{1-T_i^{\text{Unc}}}{1-e^{\text{Unc}}(X_i^{\text{Unc}})}\right)Y_i^{\text{Unc}} - \tau(X_i^{\text{Unc}})$. Note that $u_i^{\text{Unc}} = \tau(X_i^{\text{Unc}}) - \omega(X_i^{\text{Unc}}) + b_i^{\text{Unc}} - a_i^{\text{Unc}}$, which by condition 3 we can write as $u_i^{\text{Unc}} = \theta^\top X_i^{\text{Unc}} + b_i^{\text{Unc}} - a_i^{\text{Unc}}$. Hence, we have

$$\hat{\theta} - \theta = (\mathbf{X}^{\text{Unc}}(\mathbf{X}^{\text{Unc}})^\top)^{-1}(\mathbf{X}^{\text{Unc}})^\top \mathbf{u} - \theta$$
$$= ((\frac{1}{n^{\text{Unc}}}\mathbf{X}^{\text{Unc}}(\mathbf{X}^{\text{Unc}})^\top)^{-1}(\frac{1}{n^{\text{Unc}}}(\mathbf{X}^{\text{Unc}})^\top \mathbf{X}^{\text{Unc}}) - I)\theta$$
$$- (\frac{1}{n^{\text{Unc}}}\mathbf{X}^{\text{Unc}}(\mathbf{X}^{\text{Unc}})^\top)^{-1}(\frac{1}{n^{\text{Unc}}}(\mathbf{X}^{\text{Unc}})^\top \mathbf{a}^{\text{Unc}})$$
$$+ (\frac{1}{n^{\text{Unc}}}\mathbf{X}^{\text{Unc}}(\mathbf{X}^{\text{Unc}})^\top)^{-1}(\frac{1}{n^{\text{Unc}}}(\mathbf{X}^{\text{Unc}})^\top \mathbf{b}^{\text{Unc}})$$

Let $M = \mathbb{E}[XX^\top \mid E^{\text{Unc}}]$. By condition 5, we have that $\left\|\frac{1}{n^{\text{Unc}}}\mathbf{X}^{\text{Unc}}(\mathbf{X}^{\text{Unc}})^\top - M\right\|_F^2 = O_p(1/n)$. By condition 4, $\left\|(\frac{1}{n^{\text{Unc}}}\mathbf{X}^{\text{Unc}}(\mathbf{X}^{\text{Unc}})^\top)^{-1}(\frac{1}{n^{\text{Unc}}}(\mathbf{X}^{\text{Unc}})^\top \mathbf{X}) - I\right\|_F^2 = O_p(1/n)$.

Next, consider the second term:

$$\frac{1}{n^{\text{Unc}}}(\mathbf{X}^{\text{Unc}})^\top \mathbf{a}^{\text{Unc}} = \frac{1}{n^{\text{Unc}}}\sum_{i=1}^{n}(\omega(X_i^{\text{Unc}}) - \hat{\omega}(X_i^{\text{Unc}}))X_i^{\text{Unc}}$$

By Cauchy-Schwartz and condition 5,

$$\mathbb{E}[\|(\omega(X_i^{\text{Unc}}) - \hat{\omega}(X_i^{\text{Unc}}))X_i^{\text{Unc}}\|_2^2]^2 \le \mathbb{E}[(\omega(X_i^{\text{Unc}}) - \hat{\omega}(X_i^{\text{Unc}}))^4]\mathbb{E}[\|X_i^{\text{Unc}}\|_2^4] < \infty.$$

And again by Cauchy-Schwartz,

$$\|\mathbb{E}[(\omega(X_i^{\text{Unc}}) - \hat{\omega}(X_i^{\text{Unc}}))X_i^{\text{Unc}}]\|_2^2 \le \mathbb{E}[(\omega(X) - \hat{\omega}(X))^2 \mid E^{\text{Unc}}]\mathbb{E}[\|X\|_2^2 \mid E^{\text{Unc}}].$$

Conditions 1 and 2 give that $\mathbb{E}[(\hat{\omega}(X) - \omega(X))^2 \mid E^{\text{Unc}}] = O(r(n))$. Hence, by above finiteness of second moment,

$$\|(\frac{1}{n^{\text{Unc}}}\mathbf{X}^{\text{Unc}}(\mathbf{X}^{\text{Unc}})^\top)^{-1}(\frac{1}{n^{\text{Unc}}}(\mathbf{X}^{\text{Unc}})^\top \mathbf{a}^{\text{Unc}})\|_2^2 = O_p(r(n) + 1/n).$$

Finally, consider the third term:

$$\frac{1}{n^{\text{Unc}}}(\mathbf{X}^{\text{Unc}})^\top \mathbf{b}^{\text{Unc}} = \frac{1}{n^{\text{Unc}}}\sum_{i=1}^{n} b_i^{\text{Unc}} X_i^{\text{Unc}}$$

First note that $\mathbb{E}[b_i^{\text{Unc}} \mid X_i^{\text{Unc}}] = 0$ by the outcome weighting formula and hence $\mathbb{E}[b_i^{\text{Unc}} X_i^{\text{Unc}}] = 0$. By Cauchy-Schwartz and conditions 5 and 6, we have

$$\mathbb{E}[\|b_i^{\text{Unc}} X_i^{\text{Unc}}\|_2^2]^2 \le \mathbb{E}[(b_i^{\text{Unc}})^4]\mathbb{E}[\|X_i^{\text{Unc}}\|_2^4] < \infty$$

and hence

$$\|(\frac{1}{n^{\text{Unc}}}\mathbf{X}^{\text{Unc}}(\mathbf{X}^{\text{Unc}})^\top)^{-1}(\frac{1}{n^{\text{Unc}}}(\mathbf{X}^{\text{Unc}})^\top \mathbf{b}^{\text{Unc}})\|_2^2 = O_p(1/n).$$

We conclude that $\|\hat{\theta} - \theta_0\|_2^2 = O_p(r(n) + 1/n)$. Since condition 1 implies that $((\hat{\omega}(X) - \omega(X))^2 \mid E^{\text{Conf}}) = O_p(r(n))$ by Markov's theorem, we also conclude that $((\hat{\tau}(X) - \tau(X))^2 \mid E^{\text{Conf}}) = O_p(r(n) + 1/n)$. $\square$