[Reviews · NeurIPS 2018]

Reviewer 1



This paper tackles the problem of prediction of individual treatment effects given observed data and a small randomized trial. The unconfounded data can be used to estimate the effect, but the variance will be high, e.g., with importance sampling. So the authors reduce variance by including an estimate from the observed data, which no longer needs to be unbiased. I think the paper introduces an innovative approach that can be useful whenever unconfounded data is available. The experimental section is encouraging, except that I would have liked to see at least one other real-world dataset to show generalizability, and a comparison with baselines that they mention in section 3 (importance sampling and transportability ). The catch, however, is that the effect of confounders can be expressed parameterically (linear in the current paper). I think this is a reasonable assumption, but the authors make a stronger claim: "strictly weaker than other assumptions". I suggest that the authors present a formal argument for this. The current justification in Section 4 is vague. The other suggestion I have is in terms of expressing the main idea. It seems that the method is really a variance reduction trick---where a low-variance but biased estimate from the confounded data is used to reduce variance on the high-variance unbiased estimate from the confounded data. Maybe the authors can discuss this interpretation, and how this could lead to other derivative methods from this general insight?

Reviewer 2



Summary: This paper proposes a new method for leveraging experimental data to remove the bias arising from potential confounders in a related observational study. I find the key idea of the paper to be simple but clever, interesting, and potentially useful. The simulation based on real data (Section 5.2) is in my opinion quite convincing. On the other hand, I find the general exposition to be imprecise and difficult to read. I think that this manuscript has the substance to be a good paper if major work is undertaken to tighten it up and clarify the exposition. Specific comments below: Substance: 1) I believe that the method you propose is sensible, but I am not yet convinced by its practical significance. It would be helpful if you could point to a number of real world datasets on which such a method could be used. I am not asking you to actually apply your method to other datasets, nor am I asking you to point to datasets you have access to — what I am asking is whether there are any concrete datasets you know of that could benefit from your method? 2) There is a general sense in which when estimating causal effects in observational studies, bias is often more of an issue than variance. I agree with that view, overall. Still, it would be useful if you could briefly discuss the variance of your estimator, and propose a way to estimate that variance. Proving a CLT (or if you can’t prove it, at least argue heuristically for it) would be also helpful for obtaining confidence intervals, etc… 3) The precision of your estimator $\hat{\theta}$ is limited by the sample size of the UNCONF dataset. Now if you look at the mean squared error of the estimator $\hat{\tau}$ that you construct, it is quite possible that because of the potential large variance of $\hat{\theta}$, it exceeds that of a naive estimator that will be biased due to confounding but will have much lower variance thanks to the large sample size of CONF. In effect, I am pointing to the fact that there is a bias / variance tradeoff at play here, in the choice between your estimator, and a naive estimator that assumes unconfoundedness. It would be useful to discuss this point. 4) In the introduction you write: “We discuss below why this [parametric correction] assumption is possibly reasonable […]”. I think that the insight from your paper — pushing the burden of extrapolation to the confounding term — is a very good one. You discuss this briefly on lines 156-159, but I think that you should emphasize this more. Clarity: I would usually include these remarks as minor comments, but in this particular case they are not minor. I am taking the time to be specific because I really believe that the paper could be improved dramatically if the writing was tighter. 1) The language you use is very imprecise: - l.68: what do you mean by “scale” and “scope”? - l.69: you talk about unconfoundedness, but you only define it on l.91. The sentence on l. 90: “the key assumption we make about the unconfounded data is its unconfoundedness” should alert you to the fact that something is wrong. - The example in l.110-119 is confusing. What you assume is that dutiful patients are more likely to seek treatment. You should be explicit about this. As an aside, it is a matter of personal preference but you could consider wrapping this into an “example” environment to set it apart from the main text. - l.131: \omega is not an estimator, and unbiased is not the right word for it. - l 132: “[…] which interpolates between \hat{omega} to the RCT outcomes” what does this sentence mean? - l. 238-242 is confusing. Why do you superscript Y by CATE? - l. 248: you mean that you are sampling a *fraction* q’ of the rural… etc…. Please go over your paper carefully eliminating these kinds of things. 2) l.139-144 should be a lemma. The proof does not add to the comprehension. 3) Please consider not using superscripts like “Conf” and “Unc”. Single letters “C” and “U” would do. Also, using the letter E is usually not a good idea when you take expectations (even if you use a slightly different symbol for expectations). 4) Typos: - l.40: “this observational sample which might suffer […]” -> remove the word “which”. - l.131 and l.132: its \omega not omega - l.138: “doing so would be difficuly” -> difficult - l 139: “[…] relies on a simply identity” -> simple etc…

Reviewer 3



This paper tackles the challenging problem of controlling for unconfounded bias by using a smaller, but still related randomized experiment. More specifically, the authors propose an interesting methodology that first estimates the difference in conditional average outcomes in the observational data, and then a "correction" term obtained from the randomized experiment. Many authors and have studied this problem; however, few have proposed such a solution. The paper adds a new perspective that has many potential applications. The technical content of the paper appears to be correct in the sense that given their assumptions their technical results hold. My primary concern in this area is the validity of their assumptions. In particular, for the randomized experiment to be useful, they require that the relationship between any unobserved confounders and the outcome remain constant in both the observational study and the randomized experiment. I think this assumption needs a little more discussion as there are examples where it does not hold. Can you say anything more about this? I think the data generating process in their simulation study is rather strange. Usually, in observational studies, we assume that the treatment is assigned as a function of the covariates, rather than the covariates are generated as a function of the treatment. The end results are pretty much the same - there is a correlation between T and U; however, the interpretation is different. Overall, I found the paper well written and reasonably clear, although there are some minor errors, the main ones are: - writing omega instead of \omega, - not formally defining the event E - line 82 "population" I assume you mean a "super population" - this is a very important distinction in causal inference! The main contribution of this paper is the method summarized in Algorithm 1 that describes how the unobserved bias can be removed and the accompanying theorem that provides theoretical garantees. This is novel work and is something that expands the toolkit available to practitioners. I believe that this is a significant contribution as it provides a new an underexplored avenue for further research. In particular, I can envision many applied areas that can utilize and expand this work.